# Comparison of Six Phenotypic Assays with Reference Methods for Assessing Colistin Resistance in Clinical Isolates of Carbapenemase-Producing Enterobacterales: Challenges and Opportunities

**DOI:** 10.3390/antibiotics11030377

**Published:** 2022-03-11

**Authors:** Annamária Főldes, Edit Székely, Septimiu Toader Voidăzan, Minodora Dobreanu

**Affiliations:** 1Department of Microbiology, Laboratory of Medical Analysis, “Dr. Constantin Opriş” County Emergency Hospital, 430031 Baia Mare, Romania; 2Doctoral School of Medicine and Pharmacy, “George Emil Palade” University of Medicine, Pharmacy, Science and Technology, 540142 Targu Mures, Romania; 3Department of Microbiology, Central Clinical Laboratory, County Emergency Clinical Hospital, 540136 Targu Mures, Romania; edit.szekely@umfst.ro; 4Department of Microbiology, “George Emil Palade” University of Medicine, Pharmacy, Science and Technology, 540142 Targu Mures, Romania; 5Department of Epidemiology, “George Emil Palade’’ University of Medicine, Pharmacy, Science and Technology, 540142 Targu Mures, Romania; septimiu.voidazan@umfst.ro; 6Department of Clinical Biochemistry, Central Clinical Laboratory, County Emergency Clinical Hospital, 540136 Targu Mures, Romania; minodora.dobreanu@umfst.ro; 7Department of Laboratory Medicine, “George Emil Palade” University of Medicine, Pharmacy, Science and Technology, 540142 Targu Mures, Romania

**Keywords:** carbapenemase-producing Enterobacterales, colistin susceptibility testing, broth microdilution, colistin broth disc elution, Vitek 2 compact, rapid polymyxin NP test, Etest, ChromID colistin R agar, micronaut MIC-strip colistin, population analysis profiling

## Abstract

The global escalation of severe infections due to carbapenemase-producing Enterobacterales (CPE) isolates has prompted increased usage of parenteral colistin. Considering the reported difficulties in assessing their susceptibility to colistin, the purpose of the study was to perform a comparative evaluation of six phenotypic assays—the colistin broth disc elution (CBDE), Vitek 2 Compact (bioMérieux SA, Marcy l’Etoile, France), the Micronaut MIC-Strip Colistin (Merlin Diagnostika GMBH, Bornheim-Hensel, Germany), the gradient diffusion strip Etest (bioMérieux SA, Marcy l’Etoile, France), ChromID Colistin R Agar (COLR) (bioMérieux SA, Marcy l’Etoile, France), and the Rapid Polymyxin NP Test (ELITechGroup, Signes, France)—versus the reference method of broth microdilution (BMD). All false resistance results were further assessed using population analysis profiling (PAP). Ninety-two nonrepetitive clinical CPE strains collected from two hospitals were evaluated. The BMD confirmed 36 (39.13%) isolates susceptible to colistin. According to the BMD, the Micronaut MIC-Strip Colistin, the CBDE, and the COLR medium exhibited category agreement (CA) of 100%. In comparison with the BMD, the highest very major discrepancy (VMD) was noted for Etest (*n* = 15), and the only false resistance results were recorded for the Rapid Polymyxin NP Test (*n* = 3). Only the PAP method and the Rapid Polymyxin NP Test were able to detect heteroresistant isolates (*n* = 2). Thus, there is an urgent need to further optimize the diagnosis strategies for colistin resistance.

## 1. Introduction

The emergence and spread of diverse types of carbapenemase producers belonging to the Enterobacterales order (CPE) has been increasingly reported worldwide in recent years [1,2,3,4]. These pathogens are involved in different types of human infections, in both community and hospital settings, and frequently coexpress resistance to several classes of antibiotics that are critical in therapy [1,4,5].

*Klebsiella pneumoniae* is the most common globally mentioned CPE isolate and is mainly associated with nosocomial infections and has devastating effects on patient outcomes [1,3,6,7].

The global fight against the threat of antimicrobial resistance is the best strategy for preventing infections [8]. The potential impact of the current coronavirus disease (COVID-19) pandemic on antimicrobial resistance has not yet been established but was reported to possibly cause an escalation whose severity varies between geographic regions, different hospitals, and even distinct units of the same medical institution [9,10,11].

The management of infections due to multidrug-resistant (MDR) and extensively drug-resistant (XDR) CPE strains poses a significant challenge for health systems [7,12]. Colistin, tigecycline, aminoglycosides, and fosfomycin are considered second-line antimicrobials, which frequently express in vitro activity against some CPE isolates, but there are concerns regarding their current efficacy, the development of potential toxicity, and the rapid dissemination of resistance [4,7].

Colistin, also known as polymyxin E, is a cationic lipopeptide and bactericidal agent discovered more than half a century ago [13,14,15], but its systemic administration has largely been fallen out of favor over a substantial period of time, mainly because of its nephro- and neurotoxicity [16]. It recently regained major worldwide clinical importance as a last resort therapy for some severe infections due to MDR and XDR Gram-negative bacilli [17,18].

The complex mechanisms of action of colistin have not been entirely decoded [14,19]. The initial target is the lipopolysaccharide (LPS) of the outer membrane of Gram-negative bacteria, displacing the divalent cations Ca^2+^ and Mg^2+^ and leading to enhanced membrane permeability, loss of membrane integrity, and ultimately cell destruction [13,16,19]. Furthermore, colistin suppresses the endotoxin effect by neutralizing the lipid A component of the LPS and promotes bacterial cell injury through the development of reactive oxygen species and by suppressing enzymes whose functions are indispensable in the bacterial respiratory process [16,20].

Colistin sulfate and colistimethate sodium, which is a less toxic inactive prodrug, are the two commercially available forms of colistin for oral or topical administration and for the systemic route, respectively [13,16,19]. The optimal doses of colistimethate sodium are still a matter of debate [7,17], especially for critically ill patients receiving sustained low-efficiency dialysis and those with acute kidney injury [17].

Recent hospital outbreaks due to colistin-resistant CPE strains have occurred worldwide [15,21,22,23,24,25,26]. Acquired resistance to colistin is based mainly on diverse chromosomal mutations [14,19,27], but the extensive use of colistin both in human and veterinary health sectors has promoted the emergence and development of the plasmid-encoded mobile colistin resistance (*mcr*) genes starting from *mcr-1* up to *mcr-10*, with multiple variants [14,15,16,28,29].

Assessing colistin resistance continues to remain a challenge for microbiology laboratories [14,30,31]. Both the European Committee on Antimicrobial Susceptibility Testing (EUCAST) and the Clinical and Laboratory Standards Institute (CLSI) recommend the broth microdilution (BMD) as the reference method for susceptibility testing [32]. This standard technique is laborious, susceptible to errors, difficult to implement into routine practice, and presents limitations in detecting colistin heteroresistance; therefore, an alternative method with satisfactory performance is needed for routine testing [14,30]. Generally, the heteroresistance phenomenon has been described as the emergence of minor subpopulations with higher degrees of antimicrobial drug resistance within the dominant population of the same culture [33]. Population analysis profiling (PAP) is the gold standard method for the evaluation of heteroresistance to an antibiotic [33,34], but this technique is labor- and time-intensive [35], standard guidelines are lacking [33,36], and it requires a high consumption of materials. However, a limited number of studies have investigated the presence of colistin heteroresistance in CPE isolates.

In this context, the aim of this study was to perform a comparative evaluation of six phenotypic methods versus BMD, and all major discrepancies (MDs) (false resistance results) compared to the BMD were further assessed using the PAP method in order to obtain an improved algorithm for reliable and convenient determination of colistin-resistant CPE isolates in daily activity.

## 2. Results

### 2.1. Variety of Carbapenemases

The combination disc test was used for phenotypic confirmation of the following carbapenem-hydrolyzing enzymes: *K. pneumoniae* carbapenemase (KPC) (*n* = 41), oxacillinase-48-like (OXA-48-like) (*n* = 32), and metallo-β-lactamase (MBL) (*n* = 19). Of the 22 strains previously tested by polymerase chain reaction (PCR), 12 harbored *bla*_OXA-48-like_, 6 *bla*_KPC_, and 4 *bla*_NDM_ (New Delhi metallo-β-lactamase), in agreement with the combination disc test results.

### 2.2. Colistin Testing Results versus BMD

The BMD was used to confirm the colistin susceptibility of 36 (39.13%) strains and 56 (60.86%) isolates resistant to this antimicrobial agent. Distributions of colistin minimum inhibitory concentrations (MICs) obtained with the BMD ranged from 0.0625 to 64 mg/L (Table 1).

The 36 strains with MIC ≤ 0.5 mg/L by Vitek 2 Compact strongly correlated with the BMD results, whereas all isolates with MIC values between 1 and 2 mg/L by Vitek 2 Compact generated false susceptible results (*n* = 8) (Figure 1a).

A perfect linear correlation at the angle of 45° was achieved between the MIC of colistin determined by BMD and by colistin broth disc elution (CBDE) (Figure 1b).

The colistin MIC results obtained with the Micronaut MIC-Strip were strongly correlated with the reference MIC (Figure 1c), whereas a weak correlation was noted for strains determined to have an MIC ≥ 2 mg/L by the Etest method (Figure 1d). Distinct colonies within the ellipse of growth inhibition around the Etest strips were not observed, except in the case of the reference *Enterobacter cloacae* ATCC 13047.

The highest number of false negative results (*n* = 15) was documented for the Etest method, followed by Vitek 2 Compact (*n* = 8) (Table 2), and exclusively for *K. pneumoniae* isolates (Table 3).

No false positive or false negative results were recorded for any of the 82 strains of *Escherichia coli* and *K. pneumoniae* tested with ChromID Colistin R Agar (COLR). According to the manufacturer, only *E. coli*, *K. pneumoniae*, and *Salmonella* spp. are listed as target microorganisms for qualitative testing. However, one strain of colistin-resistant *E. cloacae* complex determined to have an MIC value of 32 mg/L by BMD developed on this medium, forming blue-green colonies. A positive pattern was also noted for the reference strain *E. cloacae* ATCC 13047.

Three false positive results were observed with the Rapid Polymyxin NP Test for one *E. cloacae* complex MBL strain A and two *K. pneumoniae* KPC isolates B and C (Table 3). All positive results, including these strains, were obvious following 2 h of incubation, with the exception of *E. cloacae* isolate A, where the color change from orange to yellow was clearly noted at 3 h but without the same turbidity in comparison with the positive control well. The test was replicated with the same results.

### 2.3. Performance of Commercial Methods in Relation to the BMD

The highest essential agreement (EA) was documented for the Micronaut MIC-Strip (85/92, 92.39%, 95% CI 73.21–95.43%), followed by Vitek 2 Compact (84/92, 91.30%, 95% CI 74.81–96.35%), and Etest (46/92, 50%, 95% CI 36.64–66.83%) (Table 2). A category agreement (CA) of >90% was obtained for all methods except the Etest (77/92, 83.69%, 95% CI 66.04–94.86%) (Table 2). Total agreement was noted in the case of the Micronaut MIC-Strip, CBDE, and COLR medium.

Very major discrepancies (VMDs) were strictly limited to Etest (15/56, 26.78%, 95% CI 14.04–44.18%) and Vitek 2 Compact (8/56, 14.28%, 95% CI 7.18–28.24%). Only Rapid Polymyxin NP Test induced MDs (3/36, 8.33%, 95% CI 1.74–24.35) (Table 2). The comparative testing results obtained for all strains in which VMDs and MDs were recorded are summarized in Table 3.

The sensitivity, specificity, positive predictive value (PPV), and negative predictive value (NPV) determined for all methods are presented in Table 2.

### 2.4. Colistin Testing Results versus PAP

*E. cloacae* complex strain A and *K. pneumoniae* isolates B and C presented subpopulations with a frequency at 10^8^ CFU/mL ranging from 4.0 × 10^−7^ to 6.6 × 10^−4^ (Table 4 and Figure 2). *E. cloacae* strain A was interpreted with homogeneous response in the susceptible range with the inhibitory colistin concentration in the PAP assay of 1 mg/L in comparison with the original MIC of 0.25 mg/L by BMD (a fourfold difference) (Table 3 and Table 4). The two heteroresistant *K. pneumoniae* isolates B and C had minor resistant subpopulations, and the colistin concentration, which suppressed the entire growth in the PAP technique, was at least 16-fold higher than the native MICs of 0.25 and 0.5 mg/L, respectively, obtained by BMD (Table 3 and Table 4). The heteroresistant phenotype of the two mutants belonging to *K. pneumoniae* isolates B and C remained stable after one week of serial passages on colistin-free agar (MIC > 64 mg/L) (Table 4).

All three strains were isolated from respiratory tract specimens of patients hospitalized in the intensive care unit (ICU) of the same medical institution. They had been treated with colistin according to reported susceptibility results with unfavorable outcomes (Table 4).

### 2.5. Reproducibility

The expected MIC targets set by EUCAST for BMD were established using reference strains *E. coli* ATCC 25922 and *E. coli* NCTC 13846 for all quantitative methods. The COLR medium, the Rapid Polymyxin NP Test, and the CBDE also showed reproducible results when they were repeatedly assessed with the reference strains. All the qualitative and quantitative methods, including the PAP assay, displayed reproducible results after testing with *E. cloacae* ATCC 13047.

## 3. Discussion

Our analysis investigated the performance of six phenotypic methods compared to the reference procedure BMD, and all false positive results were further evaluated using the PAP method in order to assess their appropriateness for susceptibility testing of CPE clinical isolates. To the best of our knowledge, this is the first study to evaluate this combination of tests on CPE strains.

In the clinical arena, colistin is a toxic agent reserved for severe infections due to MDR Gram-negative bacilli [7], and erroneous laboratory results and delays in reporting should be avoided [37]. Finding alternative, accurate, and more practical methods for colistin testing as an alternative to the laborious gold standard BMD method remains a challenge [17,30,37] so long as there are limited laboratories with extensive experience to perform the BMD assay [38].

According to the CLSI, the CBDE and agar dilution MIC methods are also acceptable for colistin susceptibility testing [39]. The CBDE procedure is a simple and affordable method that was obsolete but has recently regained relevance [37,39], and it shows notable concordance with the BMD for Enterobacterales strains, except when *E. coli* isolates harbor *mcr-1* genes [37,40].

In our investigation, the adopted interpretation of the CBDE results was compliant with the EUCAST breakpoints to allow easier comparison with the BMD. Simner et al. suggested that Enterobacterales isolates with a colistin MIC ≥ 2 µg/mL by the CBDE should be validated by BMD, and positive strains should subsequently be genotypically tested for *mcr* genes [40]. None of our isolates exhibited an MIC of exactly 2 mg/L, which is the cutoff value of EUCAST breakpoints. In our analysis, the sensitivity, specificity, and CA were 100% without VMDs or MDs for the CBDE method. These aspects are in agreement with early observations, with the exception of some VMDs reported especially for *mcr-1*-positive *E. coli* strains [40,41].

From a practical perspective, regarding the CBDE method, Humphries et al. mentioned concerns about their limited experience with only one type of cation-adjusted Mueller Hinton broth (CAMHB) (Remel, Lenexa, KS) in pre-aliquoted borosilicate tubes [41]. However, our research showed agreement with the reference method using a different medium (Becton Dickinson) in polypropylene tubes, even though the possibility of colistin adhering to plastic surfaces was mentioned in another study [40].

Previously reported data mentioned inadequate colistin performance testing results obtained with the Vitek 2 Compact according to the standard criteria: CA ≥ 90%, EA ≥ 90%, VMD ≤ 3%, and MD ≤ 3% [42,43,44,45]. Our results obtained with Vitek 2 Compact confirmed this, but with acceptable EA and CA of 91.30%, and MD of 0%. In contrast to some of the abovementioned publications that reported VMDs for Vitek 2 Compact between 26.3% [42] and 36% [44], our VMD was lower (14.28%), and all eight false negative results were noted exclusively in *K. pneumoniae* strains. Additionally, our collection included only four available isolates of *E. cloacae* complex, a problematic pathogen that was recognized to induce false susceptible results on Vitek 2 Compact and BD Phoenix semiautomated systems [45,46]. None of these isolates had MIC values close to the EUCAST breakpoints.

Colistin susceptibility was correctly identified by Vitek 2 Compact in all 36 of our isolates with MIC ≤ 0.5 mg/L. Eight resistant strains with MIC between 4 and 16 mg/L according to BMD were not correctly detected by this instrument, which indicated MICs between 1 and 2 mg/L. In many microbiology laboratories, Vitek 2 Compact represents an important instrument of rapid susceptibility testing [14,30], but with recognized disadvantages regarding underestimation of colistin-resistant Enterobacterales pathogens and impossibility of detection of colistin heteroresistance [45,47]. However, our study reveals that Vitek 2 Compact is a reliable instrument for diagnosing colistin resistance, with no false positive results recorded. Similar observations were previously reported by Pfennigwerth et al. [45].

All our colistin-resistant strains with MIC values close to the EUCAST breakpoints, for which false negative results were obtained by Vitek 2 Compact (*n* = 8), were accurately assessed using the Micronaut MIC-Strip, the COLR medium, and the Rapid Polymyxin NP Test. Etest failed to detect colistin resistance in any of these cases. The more these particular strains are included in a study, the higher the number of errors, which will contribute to modifying the antimicrobial susceptibility test evaluation results [48].

In our analysis, the Micronaut MIC-Strip was the only reliable MIC technique that fulfilled all requirements with CA and EA > 90%, and no VMD or MD. Furthermore, for this commercial method, Matuschek et al. demonstrated similar scores for EA and CA, acceptable MD, and no VMD on a limited collection of strains (*n* = 32) of *E. coli* and *K. pneumoniae* [48].

The COLR medium has been particularly designed for detection of some colistin-resistant Enterobacterales species both as a screening method for the detection of colistin-resistant strains directly from biological samples and a qualitative method used directly on bacterial cultures [49]. It contains chromogenic substrates allowing rapid color-based pre-identification of colonies [49]. In our investigation, COLR agar presented excellent performance with both a sensitivity and specificity of 100%, a complete CA with the BMD, and no VMD or MD recorded for *E. coli* and *K. pneumoniae*. Most of the studies assessed the performance of COLR agar using the screening technique [49,50]. However, in a recent report, Bala et al. demonstrated a CA of 94.3% between COLR agar used in the qualitative method and BMD on a collection of 87 characterized Enterobacterales strains [51]. They also remarked that this new chromogenic medium could be a reliable and practical alternative for the taxa recommended by the manufacturer [51].

In our research, the Rapid Polymyxin NP Test was shown to be a simple and rapid assay, with an adequate CA of 96.73%, a sensitivity of 100%, specificity of 91.67%, PPV of 94.92%, and NPV of 100%. The original in-house method, introduced by Nordmann et al. more than 5 years ago, demonstrated a sensitivity and specificity of 99.3% and 95.4%, respectively, when a large group of 200 Enterobacterales strains with different mechanisms of colistin resistance, including one isolate of *K. pneumoniae* previously characterized with colistin heteroresistance, was evaluated [52]. Additionally, in agreement with our findings, a recent study on the commercial test reported a sensitivity, specificity, PPV, and NPV of 100%, 95.9%, 98.3%, and 100%, respectively, on 339 Enterobacterales isolates, including an important proportion of particular strains with colistin MIC close to the cutoff breakpoint values [53].

The Rapid Polymyxin NP Test demonstrated the ability to detect colistin heteroresistance, as well as *mcr-1* and *mcr-2* producers [14,46,54]. On a collection of 70 *mcr*-1/*mcr*-2 producers belonging to the Enterobacterales order with distinct origins, Poirel et al. emphasized that the MICs obtained by the BMD were between 4 and 64 mg/L, and this commercial method showed excellent sensitivity and specificity [54].

Interestingly, our MD of 8.33% for the Rapid Polymyxin NP Test was noted in three strains, A, B, and C, with MIC ≤ 0.5 mg/L according to the BMD and applying the reference PAP assay confirmed that two *K. pneumoniae* isolates B and C were colistin-heteroresistant. It has been specified that traditional testing techniques, including the BMD, are not able to identify colistin heteroresistance, which can generate erroneous categorization, and in severe infections these essential findings can explain possible colistin treatment failures [30,34,35]. Moreover, these resistant subpopulations were assumed to contribute through chemical communication, transferring antibiotic resistance to protect more susceptible members [33]. Recently, Band et al., in a multicenter project conducted in the United States, demonstrated that heteroresistance to colistin among 408 CPE strains has largely remained underestimated [35].

In contrast, compared strictly to the BMD, Kon et al. indicated MD, VMD, sensitivity, and specificity of 1.8%, 21.1%, 78.9%, and 98.2%, respectively, for the Rapid Polymyxin NP Test, as well as many inconclusive color changes [55]. In our research, no VMD was observed, and the results of all repeated tests using different size inoculum were reproducible, conducted within the manufacturer’s recommended range and applying the same photometric device DensiCHEK, as is the case with the aforementioned authors [55]. However, Jayol et al. demonstrated, on a large collection of 223 Enterobacterales isolates (including 38 *mcr*-like producers and 19 heteroresistant isolates), excellent performance for this commercial kit with MD, VMD, sensitivity, and specificity of 5.1%, 1.9%, 98.1%, and 94.9%, respectively [56].

Our findings revealed some minor difficulties in the interpretation of results obtained with the Rapid Polymyxin NP Test only in the case of *E. cloacae* isolate A, but the PAP method illustrated a homogeneous response with a minor subpopulation (6.6 × 10^−4^) that responded to colistin concentrations below the breakpoints. The two heteroresistant *K. pneumoniae* isolates B and C showed a stable phenotype of resistance in the passaging study (MIC > 64 mg/L). Although a previous colistin treatment promoted the emergence of resistance to this antimicrobial agent in *K. pneumoniae* isolates recovered from patients hospitalized in the ICU [57], our study revealed that only the patient with *K. pneumoniae* isolate C had been previously exposed to prolonged therapy to colistin.

In line with several other authors, our Etest results illustrated unacceptably low values of EA and CA of 50% and 83.69%, respectively, and our EA and CA were lower in comparison with others [44,45,48]. The unreliable detection of colistin resistance using the Etest has already been indicated [14,30,32,37,40] and is explained by deficient distribution of the large molecule of colistin into agar media [37,40]. The VMDs outlined for Etest varied between 12% for Enterobacterales strains [44] and 41.5% for *K. pneumoniae* isolates [58]. Our VMD of 26.78% (15 false negative results) for this method was registered exclusively in *K. pneumoniae* strains. However, the main benefit of using Etest remains, which is the possibility to discover colistin-heteroresistant subpopulations, but this aspect is dependent on the type of Mueller Hinton medium used [47]. In the present study, heteroresistant isolates could not be identified using the Etest method.

The current research emphasizes the presence of colistin-heteroresistant subpopulations in *K. pneumoniae* KPC isolates B and C, which were erroneously categorized as colistin-susceptible by the BMD and all applied commercial methods with the exception of the Rapid Polymyxin NP Test.

In this context, clinical microbiology laboratories should select, validate, and implement diverse, accurate, and even combined methods into routine practice [30,38], especially in the case of challenging isolates with an MIC close to the breakpoints, or those with colistin heteroresistance. Moreover, the results of the test performance are influenced by the accessibility and the complex types of isolates included in studies [38,45,59].

Our promising results should be seen in the light of some limitations. This research was strictly limited to phenotypic methods, and the subjacent molecular mechanisms of colistin resistance were not explored despite the fact that it has been highlighted that these methods should be performed concomitantly with phenotypic tests for improved characterization of individual strains [30]. Additionally, colistin heteroresistance has not yet been associated with *mcr* genes [30,60,61]. Although it is fundamental to perform all tests using the same inoculum, this requirement could not be respected in our investigation because of the logistical constraints associated with the vast variety of methods that were performed by a single skilled person. Even when several tests were performed simultaneously, the inoculum was always used within 15 min of preparation. Furthermore, the purpose of our study was not focused strictly on the comparative evaluation of these phenotypic methods but rather an attempt to define an improved algorithm that can be successfully utilized in the routine diagnosis of colistin resistance.

### Future Challenges and Perspective

Future studies should provide unequivocal answers to challenges related to colistin susceptibility testing assays and to define an optimal, but more practical, testing strategy. Continuous assessments are essential to confirm the performance of the COLR medium as a qualitative technique and that of the CBDE, which is a recent method approved by the CLSI Guidelines. Furthermore, research should be oriented toward establishing the real significance of colistin heteroresistance and the impact of this phenomenon on phenotypic assays together with its possible involvement in the failure of therapies based on colistin. Definition of a harmonized consensus regarding heteroresistance and a standardized method for detection of this phenomenon is of high importance.

## 4. Materials and Methods

### 4.1. Bacterial Strains

A group of 92 nonduplicate clinical CPE isolates, including *K. pneumoniae* (*n* = 78), *Citrobacter freundii* (*n* = 6), *E. cloacae* complex (*n* = 4), and *E. coli* (*n* = 4), were tested. The strains were collected from the Dr. Constantin Opriş County Emergency Hospital Baia Mare, Romania (*n* = 60), from January 2017 to April 2021, and from Targu Mures County Emergency Clinical Hospital, Romania (*n* = 32), from January 2017 to April 2019. The two medical institutions are general acute care public hospitals with 920 and 1089 beds, respectively. The second is a teaching hospital.

Respiratory tract (*n* = 37), urine (*n* = 34), wounds (*n* = 13), blood (*n* = 7), and intravenous catheter tip (*n* = 1) were the sources of isolates. The strains were identified to species level based on conventional methods, Vitek 2 Compact (bioMérieux SA, Marcy l’Etoile, France) or API 20E strip (bioMérieux SA, Marcy l’Etoile, France). At both laboratories, routine testing for colistin susceptibility was performed using Vitek 2 Compact. Strains were selected based on variable levels of MICs as determined by Vitek 2 Compact and interpreted according to the EUCAST breakpoints [62,63,64,65,66,67]. On this instrument, 44 colistin-susceptible strains presented colistin MICs as follows: ≤0.5 mg/L (*n* = 36), 1 mg/L (*n* = 1), and 2 mg/L (*n* = 7). The remaining 48 isolates were colistin-resistant, with MICs between 4 and ≥16 mg/L. All isolates were stored at −70 °C and subcultured twice on solid medium before testing.

### 4.2. Data Collection

In the case where identification of strains showed discrepancies between the Rapid Polymyxin NP Test and BMD results, the patients’ electronic medical records were interrogated for colistin treatments administered before and after specimen collection, the wards of hospitalization, and clinical outcomes.

### 4.3. Identification of CPE Strains

Screening and phenotypic confirmation of carbapenem-hydrolyzing enzymes were accomplished for all strains using the modified carbapenem inactivation method (mCIM) [68,69] and the combination disc test (KPC, MBL, OXA- 48 Confirm kit, Rosco Diagnostica, Taastrup, Denmark), respectively. Of the total of 92 isolates included in our investigation, 22 strains were characterized by a multiplex PCR assay for the presence of carbapenemase-encoding genes (*bla*_KPC_, *bla*_NDM_, and *bla*_OXA-48-like_), as described in two other studies [70,71], according to the method used by Szekely et al. [72].

### 4.4. Detection of Colistin Resistance

Each CPE isolate was tested using commercial methods according to the manufacturers’ instructions and by CBDE versus BMD. In case of false positive results, strains were further examined in detail by application of the PAP method for assessing possible coexistence of residual colistin-heteroresistant subpopulations at baseline.

#### 4.4.1. Colistin MIC Determination

The BMD and the CBDE were carried out simultaneously starting from the same inoculum. The density of direct normal saline bacterial suspension was standardized to 0.5 McFarland using a calibrated photometric device (DensiCHEK, bioMérieux SA).

The reference BMD was accomplished with cation-adjusted BBL Mueller Hinton II Broth (CAMHB, reference 212322, Becton Dickinson, Sparks MD, USA) and colistin sulfate salt powder (reference C4461, Sigma-Aldrich, St. Louis, MO, USA) in 96-well, nontreated, U-bottom, sterile polystyrene plates (reference 734-2782, VWR International, Radnor, PA USA) in accordance with the international guidelines [32,73]. Sterile distilled water was used both as solvent and diluent for preparing stock solutions of colistin [39]. Five batch panels prepared in-house with 100 µL per well using serial twofold dilutions corresponding to a concentration range of 0.125 to 128 mg/L were stored at −70 °C before use. All isolates were tested in duplicate, and all wells contained a targeted final concentration of roughly 5 × 10^5^ CFU/mL microorganisms. A purity plate was prepared as a growth control for each tested isolate. The results were interpreted visually using a mirror after incubation at 35 ± 2 °C for 16–20 h in ambient air.

The CBDE was performed with 4 dilution tubes per isolate and 10 mL cation-adjusted BBL Mueller Hinton II Broth (reference 212322, Becton Dickinson) per polypropylene tube and colistin sulfate discs (10 μg, Thermo Fisher Scientific, Basingstoke, UK) in order to obtain final colistin concentrations of 0 (growth control), 1, 2, and 4 mg/L, respectively, as described previously [39]. The MICs were read by unaided eye after incubation at 33 to 35 °C for 16–20 h in aerobic atmosphere.

Susceptibility testing with Vitek 2 Compact (BioMérieux, Durham, NC) cards AST XN05 and AST N222 allowed the reporting of colistin MICs of ≤0.5 to ≥16 mg/L and was performed for most of the isolates in separate experiments. The susceptibility testing results were available within 18 h.

The Micronaut MIC-Strip Colistin (Merlin Diagnostika GmbH, Borhheim-Hensel, Germany), the gradient diffusion strip Etest (BioMérieux SA, Marcy l’Etoile, France) on Mueller Hinton E agar (MHE) (BioMérieux SA, Marcy l’Etoile, France), and ChromID Colistin R agar (COLR) assays (BioMérieux SA, Marcy l’Etoile, France) were conducted in parallel using the same bacterial suspension in another experimental session.

The design of the Micronaut MIC-Strip allowed testing one isolate per strip, with each strip consisting of 12 plastic wells forming a standard testing panel with dehydrated colistin. Then, 50 µL of each standardized bacterial suspension was homogenized in 11.5 mL CAMHB (Merlin Diagnostika GmbH, Borhheim-Hensel, Germany), followed by inoculation of each well with 100 µL of this prepared suspension. After incubation for 18–22 h at 35–37 °C, Micronaut-MIC strip evaluation was performed visually using a mirror.

Additionally, each standardized bacterial inoculum was swabbed onto MHE, and after a maximum of 15 min of drying of the agar surface, an Etest colistin strip was aseptically applied. After incubation at 36 °C for 16–20 h, the MIC results obtained by Etest were read by naked eye and a magnifying glass in reflected light.

The colistin MICs ranged ≤0.0625 to ≥64 mg/L for the Micronaut MIC-Strip and from ≤0.016 to ≥256 mg/L for Etest.

The reference PAP protocol was adapted from Liao et al. and Bergen et al. [60,74]. The preparation of agar plates respected the guidelines for MIC evaluation by the agar dilution method [73], and the prepared plates were refrigerated and used within 4 days of preparation. Isolates were evaluated starting from the standardized inoculum of 0.5 McFarland (approximately 10^8^ CFU/mL) prepared from overnight culture. Serial 10-fold saline dilutions ranging from 10^8^ to 10^2^ CFU/mL were subsequently performed, and 50 µL from each dilution was spread onto solid Mueller Hinton agar (MHA) Thermo Fisher Scientific, Basingstoke, UK) plates supplemented with 0, 0.5, 1, 2, 4, 8, and 16 µg/mL colistin sulfate (reference C4461, Sigma-Aldrich, St. Louis, MO, USA). The inoculation started with a growth control MHA plate containing a similar volume of sterile water instead of colistin, continued from plates with the lowest colistin concentration to plates with the highest drug concentration and, finally, a second growth control plate was used. CFUs on each plate were counted following 48 h of incubation at 36 °C. For each strain, the stability of the phenotype was determined as follows: a single colony selected from the plates with the highest colistin concentrations was passaged onto colistin-free agar for 7 consecutive days, followed by the reassessment of the MICs by BMD [33,75].

#### 4.4.2. Qualitative Phenotypic Assays

A streak plate procedure using 10 µL from each 0.5 McFarland bacterial suspension diluted 1:100 in sterile normal saline to a final concentration of 1 × 10^6^ CFU/mL was applied onto COLR medium. After overnight incubation under aerobic conditions, pink-red and blue-green colonies were suggestive of colistin-resistant *E. coli* and *K. pneumoniae*, respectively.

A standardized inoculum of 3.0–3.5 McFarland (10^9^ CFU/mL) for each strain prepared with a calibrated photometric device was homogenized with the Rapid Polymyxin NP Test medium (ELITechGroup, Signes, France). As a result of glucose metabolism, the phenolsulfonphthalein used as a pH indicator changed color from orange to yellow after 2–3 h of incubation, and indicative of a resistant strain in the presence of a defined colistin concentration of 2 mg/L. The interpretations were performed visually.

### 4.5. Interpretative Criteria

For all MIC determinations, the isolates were considered susceptible to colistin when the MIC ≤ 2 mg/L, consistent with the EUCAST breakpoints, including the CBDE [62,63,64,65,66,67].

All MIC values obtained by Etest were rounded up to the next twofold dilution step for comparative analysis.

A homogeneous response to colistin was designated when the highest inhibitory concentration according to the PAP assay was ≤4-fold of the native MIC obtained by BMD, while a difference more than 8-fold higher indicated heteroresistance to this antimicrobial agent [33].

### 4.6. Quality Controls

Tests using the Micronaut MIC-Strip, the Etest, the COLR medium, the Polymyxin NP Test, and the CBDE were performed at least 3 times with both a colistin-sensitive reference strain *E. coli* ATCC 25922 (0.25–2 mg/L, target value 0.5–1 mg/L) and a colistin-resistant strain *E. coli* NCTC 13846 (*mcr-1*-positive) (2–8 mg/L, target value 4 mg/L).

For Vitek 2 Compact, *E. coli* ATCC 25922 was used in quality control assurance on each new lot of cards, but *E. coli* NCTC 13846 was tested only on two lots of cards.

Quality control tests with *E. coli* ATCC 25922 were carried out for every new lot of API 20E strips.

In the case of the BMD, the two abovementioned reference strains were tested at least 20 times. The BMD quality control scheme was validated by concurrent testing of all these strains on every new lot of panels prepared in-house and on each working day. Additionally, the colony counts on inoculum were performed with *E. coli* ATCC 25922 [73].

All seven aforementioned methods were supplementary tested at least once with the positive heteroresistant strain *E. cloacae* ATCC 13047 (MIC of 256 mg/L).

For the mCIM and the combination disc test, *K. pneumoniae* ATCC BAA-1705 and *E. coli* ATCC 25922 were used to demonstrate positive and negative reactions, respectively. Colistin discs were examined with *Pseudomonas aeruginosa* ATCC 27853 and *E. coli* ATCC 25922.

In the case of PAP method, *E. coli* NCTC 13846 and *E. cloacae* ATCC 13047 were used as positive controls, while *E. coli* ATCC 25922 was the negative control.

A reproducible result was interpreted to be within ±1 dilution for MICs or achieving the same effect for the qualitative tests using the abovementioned reference strains.

### 4.7. Analysis of the Results

The results acquired with the commercial methods and the CBDE were compared with the reference BMD. In the case of discrepant results for each method, including the BMD, isolates were retested two or three times and the higher MIC value, or the results that appeared most often were accepted. The occurrence of skipped wells was noted only in limited situations, and those pathogens were retested in compliance with the CLSI Guidelines [73].

For the MIC determination methods, the essential agreement (EA = MICs within ±1 doubling dilution from the BMD MICs), the category agreement (CA = the same interpretation category with the BMD), the very major discrepancy (VMD) (false susceptible result), and the major discrepancy (MD) (false resistance result) were evaluated in accordance with the ISO 20776-2 standard [76]. A reliable technique was confirmed when the following criteria were met: CA ≥ 90%, EA ≥ 90%, VMD ≤ 3%, and MD ≤ 3% [76].

Each isolate tested with the PAP method was evaluated using several replicates both on the same working day and in independent experiments, and the most frequent result was recorded. The limit of detection in the PAP assay was one colony per plate (equivalent to 20 CFU/mL) [74]. The PAP was calculated for each strain by dividing the number of colonies obtained on the plates with the highest colistin concentration by the colony counts derived from the growth control plates [75].

### 4.8. Statistical Analysis

Sensitivity, specificity, PPV, and NPV were calculated for each method using the contingency tables. Microsoft Excel and the GraphPad InStat Demo State Software, version 3.06, San Diego, California, USA, were used for calculations.

### 4.9. Ethical Approval

Ethical approval for this study was obtained from the ethics committees of the Dr. Constantin Opriş County Emergency Hospital Baia Mare, Romania (reference number 14598/04.06.2019); Târgu Mureş County Emergency Clinical Hospital (reference number Ad. 14925/27.05.2019); and George Emil Palade University of Medicine, Pharmacy, Science, and Technology of Târgu Mureş, Romania (reference numbers 405/11.10.2019, 1024/13.07.2020, and 1217/18.12.2020).

## 5. Conclusions

The Rapid Polymyxin NP Test is easy to perform and offers rapid results and excellent performance compared to both the BMD and the PAP assay, including in the detection of colistin heteroresistance. The possibility of silent clonal expansion of such heteroresistant mutants in the hospital environment is of great concern, as long as the BMD and the other commercial methods used fail to detect isolates with this particular phenomenon of resistance. There is an urgent need to optimize diagnosis strategies because the reference phenotypic method PAP used for heteroresistance detection cannot feasibly be integrated into routine practice.

Strictly according to the BMD, the Micronaut MIC-Strip, the CBDE, and COLR medium exhibit the best performances in detecting colistin resistance. This report highlights the difficulties of Vitek 2 Compact in detecting isolates with MIC values close to the EUCAST breakpoints. Consequently, in situations when the reference BMD and the PAP technique cannot be performed simultaneously as confirmation, we propose an improved approach of combining all susceptible results obtained with Vitek 2 Compact, the CBDE, the COLR medium, and the Micronaut MIC-Strip with the Rapid Polymyxin NP Test. The performance of Etest gradient strips is unsatisfactory due to an unacceptable number of false negative results.

## Figures and Tables

**Figure 1 antibiotics-11-00377-f001:**
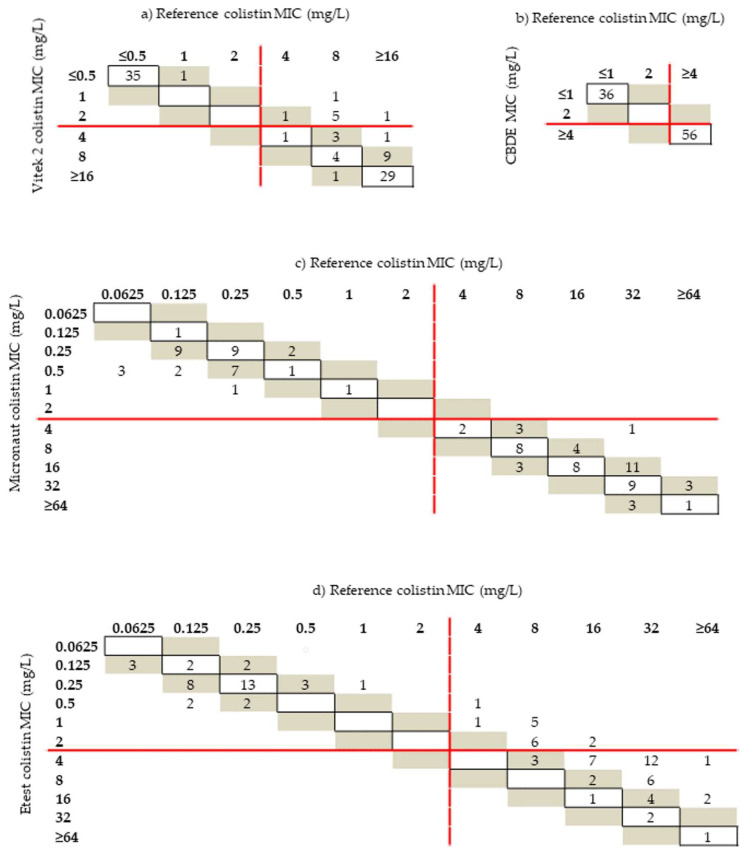
Scattergrams of correlation between reference broth microdilution (BMD) and Vitek 2 Compact (**a**), colistin broth disc elution (CBDE) (**b**), Micronaut MIC-Strip (**c**), and Etest (**d**) for all isolates. Minimum inhibitory concentrations (MICs) identical with those obtained by BMD and the essential agreement (EA) (MICs within ± 1 doubling dilution compared to the BMD) are mentioned as strain numbers within boxes and in shaded gray cells, respectively. The red lines represent the EUCAST breakpoints (≤2 mg/L indicates susceptibility).

**Figure 2 antibiotics-11-00377-f002:**
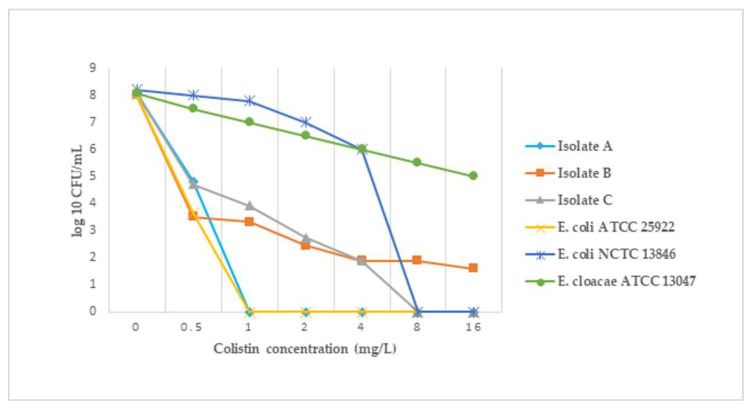
The population analysis profile of isolates A (susceptible), B, C (heteroresistant), and the reference strains at an initial inoculum of 10^8^ CFU/mL The data shown are representative of multiple replicates performed in the same experiment and on different working days for each strain.

**Table 1 antibiotics-11-00377-t001:** Distributions of colistin MICs determined by BMD for all isolates.

Species	Carbapenemase Type	Colistin Reference MIC (mg/L)										
		0.0625	0.125	0.25	0.5	1	2	4	8	16	32	≥64
*Klebsiella pneumoniae*	KPC (*n* = 41)	0	3	4	1	0	0	2	12	6	10	3
	OXA-48-like (*n* = 29)	0	4	5	1	0	0	0	2	5	11	1
	MBL (*n* = 8)	0	0	5	0	0	0	0	0	1	2	0
*Citrobacter freundii*	MBL (*n* = 6)	0	3	1	1	1	0	0	0	0	0	0
*Enterobacter cloacae* complex	MBL (*n* = 4)	2	0	1	0	0	0	0	0	0	1	0
*Escherichia coli*	OXA-48-like (*n* = 3)	1	1	1	0	0	0	0	0	0	0	0
	MBL (*n* = 1)	0	1	0	0	0	0	0	0	0	0	0
Total	*n* = 92	3	12	17	3	1	0	2	14	12	24	4

**Legend.** MICs: minimum inhibitory concentrations; BMD: reference broth microdilution; KPC: *Klebsiella pneumoniae* carbapenemase; OXA-48-like: oxacillinase-48-like; MBL: metallo-β-lactamase. The red line corresponds to the EUCAST breakpoints (≤2 mg/L indicates susceptibility).

**Table 2 antibiotics-11-00377-t002:** Performance features of Vitek 2 Compact, Micronaut MIC-Strip, Etest, COLR medium, Rapid Polymyxin NP Test, and CBDE of all isolates according to the BMD.

Parameter	Vitek 2 Compact	Micronaut MIC-Strip	Etest, MHE	COLR Medium	Rapid Polymyxin NP Test	CBDE
True positive (*n*)	48	56	41	55	56	56
False positive (*n*)	0	0	0	0	3	0
False negative (*n*)	8	0	15	0	0	0
True negative (*n*)	36	36	36	27	33	36
Total (*n*)	92	92	92	82	92	92
Sensitivity (%)	85.71	100	73.21	100	100	100
Specificity (%)	100	100	100	100	91.67	100
PPV (%)	100	100	100	100	94.92	100
NPV (%)	81.82	100	70.59	100	100	100
EA (%)	91.30	92.39	50	NA	NA	NA
CA (%)	91.30	100	83.69	100	96.73	100
VMD (%)	14.28	0	26.78	0	0	0
MD (%)	0	0	0	0	8.33	0

**Legend.** BMD: reference broth microdilution; MIC: minimum inhibitory concentration; MHE: Mueller Hinton E agar; COLR: ChromID Colistin R agar; CBDE: colistin broth disc elution; PPV: positive predictive value; NPV: negative predictive value; EA: essential agreement; CA: category agreement; VMD: very major discrepancy; MD: major discrepancy; NA: not applicable.

**Table 3 antibiotics-11-00377-t003:** The VMDs and MDs identified in all isolates in comparison with the BMD.

Strain	Carbapenemase Type	Vitek 2 Compact (MIC mg/L)	Micronaut MIC-Strip (MIC mg/L)	Etest, MHE (MIC mg/L)	COLR Medium	Rapid Polymyxin NP Test	CBDE (MIC mg/L)	BMD (MIC mg/L)
*Enterobacter* *cloacae* complex (*n* = 1) A	MBL	≤0.5	0.25	0.25	Negative	Positive ^1^	≤1	0.25
*Klebsiella pneumoniae* (*n* = 1) B	KPC	≤0.5	0.5	0.25	Negative	Positive	≤1	0.25
*K. pneumoniae* (*n* = 1) C	KPC	≤0.5	0.25	0.25	Negative	Positive	≤1	0.5
*K. pneumoniae* (*n* = 1) D	OXA-48-like	1	16	2	Positive	Positive	≥4	8
*K. pneumoniae* (*n* = 1) E	KPC	2	4	2	Positive	Positive	≥4	8
*K. pneumoniae* (*n* = 1) F	KPC	≥16	8	2	Positive	Positive	≥4	16
*K. pneumoniae* (*n* = 1) G	KPC	≥8	16	2	Positive	Positive	≥4	8
*K. pneumoniae* (*n* = 1) H	KPC	4	4	0.5	Positive	Positive	≥4	4
*K. pneumoniae* (*n* = 1) I	KPC	4	8	1	Positive	Positive	≥4	8
*K, pneumoniae* (*n* = 1) J	KPC	2	8	2	Positive	Positive	≥4	16
*K. pneumoniae* (*n* = 1) K	KPC	2	4	1	Positive	Positive	≥4	4
*K. pneumoniae* (*n* = 2) L, M	KPC	2	8	1	Positive	Positive	≥4	8
*K. pneumoniae* (*n* = 1) N	KPC	4	8	1	Positive	Positive	≥4	8
*K. pneumoniae* (*n* = 1) O	KPC	2	8	2	Positive	Positive	≥4	8
*K. pneumoniae* (*n* = 1) P	KPC	2	8	1	Positive	Positive	≥4	8
*K. pneumoniae* (*n* = 1) Q	KPC	≥8	4	2	Positive	Positive	≥4	8
*K. pneumoniae* (*n* = 1) R	KPC	8	8	2	Positive	Positive	≥4	8

**Legend.** VMDs: very major discrepancies (marked in red); MDs: major discrepancies (marked in blue); BMD: reference broth microdilution; MBL: metallo-β-lactamase; KPC: *K. pneumoniae* carbapenemase; OXA-48-like: oxacillinase-48-like; MIC: minimum inhibitory concentration; MHE: Mueller Hinton E agar; COLR: ChromID Colistin R agar; CBDE: colistin broth disc elution. ^1^ Color change detected at 3 h of incubation but without the same turbidity in comparison to the positive control.

**Table 4 antibiotics-11-00377-t004:** Clinical aspects and the PAP results of the three isolates with false positive results obtained with the Rapid Polymyxin NP Test.

Strain	Previous Colistin Therapy	Highest Colistin Concentration of Growth in PAP (mg/L)	Inhibitory Colistin Concentration in PAP (mg/L)	Frequency at Highest Colistin Concentration of Growth	MIC by BMD of Colonies before 7 Days Passages (mg/L)	MIC by BMD of Colonies after 7 Days Passages (mg/L)	Strain Classification by PAP
*E. cloacae* complex A	No	0.5	1	6.6 × 10^−4^	0.25	0.25	hO-S
K. pneumoniae B	No	16	≥32	4.0 × 10^−7^	>64	>64	hR
K. pneumoniae C	Yes ^1^	4	8	8.0 × 10^−7^	>64	>64	hR

**Legend.** PAP: population analysis profiling; MICs: minimum inhibitory concentrations; BMD: reference broth microdilution; hO-S: homogeneous response susceptible; hR: heteroresistant response. ^1^ Previous parenteral colistin treatment for 2 weeks.

## Data Availability

Data are contained within the article.

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
