# Peer review of "Comparison of Six Phenotypic Assays with Reference Methods for Assessing Colistin Resistance in Clinical Isolates of Carbapenemase-Producing Enterobacterales: Challenges and Opportunities"

_antibiotics, 2022, doi:10.3390/antibiotics11030377_

Round 1

Reviewer 1 Report

The revisions have much improved this paper and I am happy to recommend acceptance.

Author Response

We are very grateful to the Referee 1 for the precious time in reviewing our paper and offering the great opportunity to improve the article and our knowledge.

We corrected mainly the following:

  • We removed the blue colour which marked the old changes made in the manuscript and deleted the strikethrough text which was also marked in blue colour.
  • We added the explanation for the abbreviation CFU/ml at the first occururence in the main text (row 178) and in the legend of figure 2 (row 209) and deleted the term from row 434.
  • We mentioned the whole name of Pseudomonas at the first occururence in the main text (row 519).
  • We mentioned in the section 2.3 the 95% CI (rows 163-170).

All new modifications in the revised version of the manuscript have
been highlighted in yellow.

Open Review: (x) English language and style are fine/minor spell check required

Response: We will apply for a specialist English language editing service provided by MDPI when will submit the revised manuscript.

Reviewer 2 Report

Here, the authors compared the phenotypic assays (Vitek2 Compact, Micronaut MIC-Strip, Etest, COLR, Rapid Polymyxin NP test, CBDE) with reference broth microdilution for assessing Colistin resistance in Carbapenease-Producing Enterbacterales using CPE from hopsital. Performance including discrepancies (false positive/negative results etc.) were evaluated and discussed. Methods were QC-ed by commercial strains. Overall, methods and results were clearly described, well organized, clinically revelant and I recommend this article for publication.   Some addtional comments below:  
  1. It's nice/representative to have strains with different carbapenemases-producing species. In line 110, the authors should give full name of NDM, I guess it refer to New Delhi metallo-beta-lactamase 1? Are there any VIM-type strains?
  2. I recommend adding figure axies title using same font in Adobe software (PS or illustrator)
  3. Was the data shown representative from several independent experiments (biological/technical replicates)? The authors should include in figure legend.

Author Response

We are very grateful to the Referee 2 for the careful reading of the paper and for his valuable comments and constructive suggestions which helped us to improve the current version.

In addition to the modifications listed below, we corrected mainly the following:

  • We removed the blue colour which marked the old changes made in the manuscript and deleted the strikethrough text which was also marked in blue colour.
  • We added the explanation for the abbreviation CFU/ml at the first occururence in the main text (row 178) and in the legend of figure 2 (row 209) and deleted the term from row 434.
  • We mentioned the whole name of Pseudomonas at the first occururence in the main text (row 519).
  • We mentioned in the section 2.3 the 95% CI (rows 163-170).

All new modifications in the revised version of the manuscript have
been highlighted in yellow.

Open Review: (x) English language and style are fine/minor spell check required

Response: We will apply for a specialist English language editing service provided by MDPI when will submit the revised manuscript.

Point 1: It's nice/representative to have strains with different carbapenemases-producing species. In line 110, the authors should give full name of NDM, I guess it refer to New Delhi metallo-beta-lactamase 1? Are there any VIM-type strains?

Response 1: Thank you for this observation. We have added the full name of NDM in the line 110. Out of the 22 strains previously characterized by a multiplex PCR assay no VIM-type isolate was identified.

Point 2: I recommend adding figure axies title using same font in Adobe software (PS or illustrator)

Response 2: Thank you for this suggestion. In figure 2 we have modified the font of axis titles to be concordant to the rest of the text.

Point 3: Was the data shown representative from several independent experiments (biological/technical replicates)? The authors should include in figure legend.

Response 3: Thank you for pointing this out. We have modified the section Material and methods, 4.4.1. and we have deleted the term “in triplicate” at rows 468. We have completed the section 4.7 and have added the following phrase at rows 538-540: “Each isolate tested with the PAP method was evaluated in several replicates both in the same working day and in independent experiments and the most frequent result was recorded”. We have also included in the legend of figure 2 that the results were representative from multiple replicates performed both in the same experiment and in different working days for each strain.

This manuscript is a resubmission of an earlier submission. The following is a list of the peer review reports and author responses from that submission.

Round 1

Reviewer 1 Report

This work compares different techniques for identification of colistin resistance in  Carbapenem-resistant Enterobacterales. Globally, the article reports nicely-performed experiments and results

A few points shall be modified

  • please detail the Micronaut technique. Many readers may not be familiar with that specific technique
  • Underline the TAT of each technique
  • For all strains that may be problematic, perform whole genome sequencing to be sure that those strains do not harbor any colistin resistance mechanism that has not be identified using phenotypic techniques
  • For all problematic strains , n search for colistin heteroresistance. If heteroresistance is identified, then reclassified those strains as true resistant strains !!! From a clinical point of view this is very important point since colistin treatment may fail in those cases.

Reviewer 2 Report

The paper looks at a variety of testing methods for the determination of MIC for colistin.

Overall the paper is excellent and I recommend acceptance subject to some minor comments and corrections

Were the isolates screened for mcr genes? If not an a short explanation should be added to the text as to why.

The authors state "Carbapenemase-encoding genes (blaKPC, blaNDM, and blaOXA-48-like) were confirmed in 22 isolates tested by a multiplex PCR technique, as described previously [41]."

Please add this to your materials and methods.
